# High-Glass-Transition Polyesters Produced with Phthalic Anhydride and Epoxides by Ring-Opening Copolymerization (ROCOP)

**DOI:** 10.3390/polym15132801

**Published:** 2023-06-24

**Authors:** Selena Silvano, Matteo Proverbio, Adriano Vignali, Fabio Bertini, Laura Boggioni

**Affiliations:** 1Institute of Chemical Science and Technologies—“G. Natta”, National Research Council, Via A. Corti 12, 20133 Milan, Italy; selena.silvano@scitec.cnr.it (S.S.); matteo.proverbio@outlook.it (M.P.); adriano.vignali@scitec.cnr.it (A.V.); 2National Interuniversity Consortium of Materials Science and Technology, INSTM, Via Giuseppe Giusti 9, 50121 Firenze, Italy

**Keywords:** polyesters, salen-type complexes, ring-opening copolymerization (ROCOP), epoxides, cyclic anhydrides, high glass transition temperature

## Abstract

Polyesters with a high glass transition temperature above 130 °C were obtained from limonene oxide (LO) or vinylcyclohexene oxide (VCHO) and phthalic anhydride (PA) in the presence of commercial salen-type complexes with different metals—Cr, Al, and Mn—as catalysts in combination with 4-(dimethylamino) pyridine (DMAP), bis-(triphenylphosphorydine) ammonium chloride (PPNCl), and bis-(triphenylphosphoranylidene)ammonium azide (PPNN_3_) as cocatalysts via alternating ring-opening copolymerization (ROCOP). The effects of the time of precontact between the catalyst and cocatalyst and the polymerization time on the productivity, molar mass (*M*_w_), and glass transition temperature (*T*_g_) were evaluated. The polyesters were characterized by a molar mass (*M*_w_) of up to 14.0 kg/mol, a narrow dispersity *T*_g_ of up to 136 °C, and low (<3 mol%) polyether units. For poly(LO-*alt*-PA) copolymers, biodegradation tests were performed according to ISO 14851 using the respirometric biochemical oxygen demand method. Moreover, the vinyl double bond present in the poly(LO-*alt*-PA) copolymer chain was functionalized using three different thiols, methyl-3-mercaptopropionate, isooctyl-3-mercaptopropionate, and butyl-3-mercaptopropionate, via a click chemistry reaction. The thermal properties of poly(LO-*alt*-PA), poly(VCHO-*alt*-PA) and thiol-modified poly(LO-*alt*-PA) copolymers were extensively studied by DSC and TGA. Some preliminary compression molding tests were also conducted.

## 1. Introduction

The conversion of natural resources into value-added polymers is gaining momentum due to environmental issues, forecasted depletion of fossil resources, and stricter waste legislation [1,2]. Recently, the attention has been shifting toward the use of renewable and/or bio-based monomers (epoxides) as a way to produce polymers with an improved sustainability footprint. The combination of bio-based and synthetic building blocks is a promising strategy for the synthesis of advanced materials.

Polyesters are important as degradable, sustainable polymers that can contribute to a circular plastic economy. Many monomers are bio-derived, and many polyesters are recyclable because the ester linkages can undergo chemical recycling and/or degradation. Amorphous, high-glass-transition-temperature (*T*_g_ > 100 °C), bio-derived polyesters are particularly sought for use as thermoplastics and, within block polymer structures, as pressure-sensitive adhesives or thermoplastic elastomers [3,4,5,6].

In this respect, the use of terpene-derived epoxides has offered new potential for bio-derived polyesters [7,8,9,10,11] and polycarbonates [12,13,14,15,16], displaying similar or even improved properties compared with conventional fossil-source-derived polymers. In particular, commercially available and inexpensive limonene oxide (LO), the epoxidized form of the highly abundant terpene limonene, is an attractive, readily available, bifunctional, and renewable monomer; its use in polyester production is steadily increasing [17,18]. This increase is mainly due to the development of the ring-opening copolymerization (ROCOP) (Figure 1) of epoxides and cyclic anhydrides, as an alternative polyester chain-growth route with respect to traditional lactone ring-opening polymerization (ROP) [19,20,21].

Although ROP is an excellent controlled polymerization route, ROCOP represents an opportunity to broaden the range of materials produced and to overcome some of the limitations of ROP. In particular, the properties of the produced materials, including thermal properties (glass transition and thermal decomposition temperatures), can be simply controlled by changing the epoxide or the cyclic anhydride. One of the most commonly used monomer is the semi-aromatic phthalic anhydride (PA). PA, in the presence of monocyclic epoxides such as propylene oxide, yield materials with *T*_g_ values of 40–70 °C; when 1,4-cyclohexadiene oxide or cyclohexene oxide (CHO) is employed, *T*_g_ reaches values of 128 °C (*M*_n_ = 7.5 kg/mol) [22] and 147 °C (*M*_n_ = 16.3 kg/mol) [23], respectively.

The initial report from Coates et al. included the copolymerization of LO with diglycolic anhydride, catalyzed by an efficient single-site β-diiminate zinc catalyst, leading to a material with a molar mass of 36 kg/mol and *T*_g_ of 51 °C [24]. In 2011, Thomas reported a strategy based on tandem catalysis to produce alternating polyesters from renewable materials. Herein, salen-type complexes with chromium, aluminum, and cobalt as metals were used as efficient catalysts for the cyclization of dicarboxylic acids, followed by the alternating copolymerization of the resulting anhydrides with epoxides. The aluminum derivative was the most active in LO/camphor anhydride copolymerization, with a comonomer conversion of 85%. By properly tuning the anhydride/catalyst ratio, molar mass up to 27 kg/mol was obtained [25].

In 2013, Duchateau et al. synthesized partially bio-based alternating polyesters from LO and phthalic anhydride by a series of metal *t*-Bu–salophen complexes combined with different cocatalysts [26]. The catalytic system works only in bulk, and chromium-chloride-based catalysts provide the best results without any major decrease in catalytic activity. The influence of the different cocatalysts on molar mass and activity was studied, and a trend in which PPN^+^Cl^−^ showed the highest activity in the copolymerization, followed by DMAP was found. The molar masses were 9.7 and 7.2 kg/mol, respectively. The *T*_g_ for polyesters with DMAP as the cocatalyst ranged from 29 to 82 °C, with the higher value having a molar mass of 7.2 kg/mol. Recently Kleij et al. presented an Fe(III)-based catalyst derived from an aminotriphenolate ligand, which, under mild conditions and with PPNCl as the initiator, produced semi-aromatic polyesters from various terpene-based monomers, including *cis/trans*-LO, *cis*-LO, limonene dioxide, and different aromatic anhydrides [27]. The best result was obtained with *cis*-LO, producing a poly(LO-*alt*-PA) with an appreciable *T*_g_ of 141 °C and a molar mass of 16.4 kg/mol. The use of a more rigid and cyclic anhydride such as 1,8-naphthalic anhydride afforded a copolymer with *T*_g_ values up to 243 °C, even if the comonomer conversion was approximately around 50% and the molar mass was low (*M*_n_ = 2.2 kg/mol). In 2017, Mazzeo et al. achieved the synthesis of polyester from cyclohexene oxide (CHO) polymerized with different anhydrides, such as succinic anhydride and maleic anhydride in bulk, as well as in toluene solution, using bimetallic aluminum–alkyl complexes in the presence of different cocatalysts (e.g., DMAP, PPNCl, or TBABr) and reaching an *M*_n_ of only up to 5.5 kg/mol, but no data about the *T*_g_ were reported [28]. Successively, they reported the synthesis of two dinuclear aluminum complexes bearing dinaphthalene bridging Schiff bases as catalysts in the ROP of cyclic esters such as *rac*-lactide (*rac*-LA) and ε-caprolactone (ε-CL) and in the ROCOP of PA with CHO and LO [29]. The polymerization of vinylcyclohexene oxide (VCHO) with PA was also achieved using amine-bis(phenolate) chromium(III) complexes [30] as well as chromium complexes with an [OSSO]-type bis(phenolato) dianionic ligand in the presence of DMAP as a cocatalyst [31]. In 2022, Ward et al. reported the copolymerization of CHO or VCHO and PA using metallocene catalysts, with polymerization performance in terms of molecular masses and conversion of PA being somewhat modest in comparison with that achieved using DMAP alone [32]. Recently, Urban [33] reported advances in stimuli-responsive commodity polymers obtained from functionalized polyesters with cleavable bonds, such as disulfide, thioketal, or β-selenylated carbonyl linkages.

Nowadays, the synthesis of polyesters with a high *T*_g_ and an appreciable molar mass by using low-cost commercial catalysts such as the salen-type complexes is still a challenge in the synthesis of bio-based polymeric materials. In this study, we explored the synthesis of semi-aromatic poly(LO-*alt*-PA) as well as of poly(VCHO-*alt*-PA), as reported in Figure 1, with a series of commercial complexes.

The obtained polymers were characterized by size exclusion chromatography (SEC), differential scanning calorimetry (DSC), thermogravimetric analysis (TGA), and nuclear magnetic resonance (^1^H-NMR). This study highlights the key role of the pre-contact step between catalyst and co-catalyst in obtaining polyesters with a molar mass of up to 14 kg/mol and a glass transition temperature of up to 134 °C in high yields, providing a route for a possible industrial scale-up for bio-based polyesters. Moreover, the biodegradability data of selected poly(LO-*alt*-PA) copolymers are reported. Finally, the functionalization of the vinyl double bonds present in the poly(LO-*alt*-PA) copolymer chain using different thiols by a click chemistry reaction is presented.

## 2. Materials and Methods

### 2.1. Materials

(R-R)-*N,N′*-bis (3,5-di-*tert*-butylsalicylidene)-1,2-cyclohexanediaminochromium(III) chloride, (R-R)-*N,N′*-bis (3,5-di-*tert*-butylsalicylidene)-1,2-cyclohexanediaminomanganese(III) chloride, methyl-3-mercaptopropionate (M3MP), *iso*octyl-3-mercaptopropionate (I83MP), butyl-3-mercaptopropionate (B3MP), and azobis(2-methylpropionitrile) (AIBN) were purchased from Sigma-Aldrich (Milan, Italy) and used as received. (R-R)-*N,N′*-bis(3,5-di-*tert*-butylsalicylidene)-1,2-cyclohexanediaminoaluminum chloride was purchased from Strem Chemicals (Newburyport, MA, USA) and used as received. Limonene oxide and dichloromethane were purchased from Sigma-Aldrich (Milan, Italy) and distilled over CaH_2_, and stored on 4 Å molecular sieves under nitrogen. Toluene was purchased from Sigma-Aldrich (Milan, Italy) distilled over sodium and stored under nitrogen. Phthalic anhydride was purchased from Sigma-Aldrich (Milan, Italy) and recrystallized from dichloromethane prior to use. Bis-(triphenylphosphorydine) ammonium chloride (PPNCl) was dissolved in dichloromethane and precipitated in diethyl ether twice. Then, 4-(dimethylamino) pyridine (DMAP) was double recrystallized from toluene. The cocatalyst bis-(triphenylphosphorydine) ammonium azide (PPNN_3_) was synthesized following procedures described in the literature [34]. AIBN was recrystallized twice from methanol prior to use. All manipulations were performed under an inert atmosphere or in a nitrogen-filled MBraun (MBRAUN INERTGAS-System, GMBH, Garching, Germany) glove box unless otherwise stated.

### 2.2. Polymerization in Solvent with Precontact Step

In a glove box, in a 10 mL crimp cap vial equipped with a stirring bar, a mixture of catalyst and cocatalyst was charged in the presence of 1 mL of toluene, which was kept under stirring for 1 h or 24 h (precontact step). Then, the epoxide and anhydride were added. The ratio between catalyst: cocatalyst: epoxide: anhydride was 1:1:250:250. Then, the vial was placed in an aluminum heating block mounted on top of a stirrer/heating plate. At the end of the polymerization, the crude product was precipitated twice in methanol and collected after filtration through a 0.45 µm nylon filter. All the analyses were performed on purified samples.
$$\mathrm{Yield}\;\left(\%\right)\;\mathrm{was}\;\mathrm{calculated}\;\mathrm{as}\;\frac{yield(g)}{Oxirane(g)+Anhydride(g)}\times 100$$

### 2.3. Preparation of LOPASH through Functionalization of Poly(LO-alt-PA) with Thiols

*Poly(LO-alt-PA)* was dissolved in dry dichloromethane to produce a 2 wt % solution. We added 28 eq of thiol and 0.3 eq of AIBN to the polymeric solution. The solution was kept at reflux for 24 h under stirring. Then, it was concentrated and precipitated in methanol and collected after filtration [35].

### 2.4. Methods

The copolymers were weighted in a 5 mm NMR tube and dissolved in CHCl_3_. The spectra were recorded on a Bruker (Billerica, MA, USA) Avance 400 instrument (400 MHz (^1^H); 100.58 MHz (^13^C); pulse angle = 12.50 ms; acquisition time = 0.94 s; delay = 16 s). The probe head was pre-equilibrated at a fixed temperature of 35 °C.

Differential scanning calorimetry (DSC) analysis was performed on a Perkin Elmer (Waltham, MA, USA) DSC 8000 instrument. The measurement was carried out from 0 to 180 °C using heating and cooling rates of 20 °C/min. The glass transition temperature *T*_g_ was recorded during the second heating scan.

Molar mass analysis was performed using approximately 12 mg of polymer in THF at 35 °C with a size exclusion chromatography (SEC) system from Waters W600 (Millford, MA, USA), equipped with a differential refractometer (Waters 410). The column set was composed of three columns (Polypore, Oligopore, 50 Å) from Polymer Laboratories (Church Stretton, UK). The SEC system was calibrated using polystyrene standards.

Thermogravimetric analysis (TGA) was performed on a PerkinElmer (Waltham, MA, USA) TGA 7 instrument at a scan rate of 20 °C/min under a nitrogen atmosphere. TGA and derivate thermogravimetry (DTG) curves were recorded from 50 to 750 °C.

Biodegradability was determined by the respirometric method according to ISO 14851:1999 [36]. Biochemical oxygen demand (BOD) in a closed respirometer was measured with an OxiTop^®^ system. Selected LOPA and CHOPA samples, and microcrystalline cellulose, as the positive reference, were tested. All the samples were introduced in amber bottles together with 164 mL of test medium under magnetic stirring and placed in an incubator at a constant temperature of 50 °C for 60 days. The medium consisted of salts dissolved in water and an appropriate volume of inoculum. The salts guaranteed the correct amount of nutrients for the microorganisms and the maintenance of pH at a value of 7.4; the inoculum was obtained by filtering the supernatant of a suspension of 10 g of mature compost in 100 mL of test medium. The degree of biodegradation was calculated according to Equation (1):(1)$$\mathrm{Biodegradation}\;(\%)=\frac{{\mathrm{BOD}}_{s}\times 100}{\mathrm{ThOD}}$$
where *ThOD* is the theoretical oxygen demand of the test material, calculated as the oxygen demand found by establishing the relationship between oxygen and other chemical substances that are oxidizable in the chemical reaction [37]; ${\mathrm{BOD}}_{s}$ is the specific BOD of the test material, which is determined according to Equation (2):(2)$${\mathrm{BOD}}_{s}=\frac{{\mathrm{BOD}}_{t}-{\mathrm{BOD}}_{b}}{\rho_{t}}$$
where ${\mathrm{BOD}}_{t}$ is the BOD of the flask containing the test material, ${\mathrm{BOD}}_{b}$ is the BOD of the blank flask, and *ρ_t_
*is the concentration of the test material in the flask.

## 3. Results and Discussion

### 3.1. ROCOP of Limonene Oxide and Phthalic Anhydride

Limonene oxide is a bio-renewable monomer obtained from non-food materials and derived from limonene, a naturally occurring terpene that is available in large amounts in the peel of many citrus fruits. Considering the structural analogies between cyclohexene oxide (CHO) and LO and taking advantage of the results presented in a previous study [23] on the influence of the polymerization conditions on both the molar mass and glass transition temperature in the synthesis of poly(CHO-*co*-PA), we explored the reactivity of the mixture of *cis-* and *trans*-R-LO, the commercially available monomer, in the ROCOP with PA. A series of limonene oxide copolymerizations with phthalic anhydride as the comonomer were performed by the salen-type complexes centered with three different metals, Cr (**1**), Al (**2**), and Mn (**3**), in combination with DMAP, PPNCl, and PPNN_3_ as cocatalysts, which are the most efficient cocatalysts for ROCOP (Figure 1) [37]. To avoid any protonated impurities that may shorten the polymer chain length by acting as a chain-transfer agent, LO, PA, the cocatalysts and toluene were purified or distilled prior to polymerization as described in Section 2.1 and all experiments were performed under an inert atmosphere [34,38].

Based on the screening conducted with cyclohexene oxide and phthalic anhydride [23], the copolymerization reactions of LO with the PA were performed using 1 h of precontact step between the catalyst and cocatalyst and different polymerization times of 3, 24, and 48 h. In addition, the ROCOP of LO at 18 h was performed using a precontact time of 24 h (Table 1). The polyesters obtained from the ROCOP of LO and PA, poly(LO-*alt*-PA), are listed in Table 1 as LOPA, and the numbers correspond to the sequential number of experiments performed. In Appendix A, a typical SEC elution trace of poly(LO-*alt*-PA) is reported.

In contrast to ROCOP in the presence of CHO, where *M*_n_ of 15 kg/mol and *T*_g_ of 147 °C [23] are obtained at 3 h of polymerization time, the yields of copolymers obtained in the presence of LO were low (LOPA 15, Table 1) or even non existent (LOPA 07 and LOPA 11, Table 1) with the investigated catalysts and DMAP as the cocatalyst. For polymerization times of 24 and 48 h, the yield increased with the polymerization time, reaching a yield greater than 84% with the three catalysts and DMAP at 48 h. Molar masses linearly increased with increasing reaction time. For a polymerization time of 48 h with 1 h of precontact step, the highest molar masses were obtained with all catalysts (**1**–**3**, Figure 2). Molar masses higher than 10 kg/mol were also obtained at a shorter time (18 h) but with a longer precontact step (24 h) (Figure 2).

The PPNCl cocatalyst in the presence of catalysts **1**–**3** was active at 3 h of polymerization, reaching a yield of 20% with **1** (LOPA 05, Table 1) and showing a behavior similar to DMAP in terms of yield, molar mass, and glass transition temperature (Figure 2). For DMAP and PPNCl, molar masses above 12.5 kg/mol were also obtained with a shorter polymerization time (18 h) but with a precontact time of 24 h (Figure 2).

From the comparison of the polymerization for different precontact times of 1 and 24 h, we found that the time spent in the precontact step positively affected the polymerization yield and molar mass of polyesters using **1**–**3** with DMAP and PPNCl as the cocatalyst. For both DMAP and PPNCl cocatalysts, as the reaction time increased, the *M*_w_ gradually increased with a narrow molar mass distribution (*M*_w_*/M*_n_ ca. 1.2), indicating a living polymerization behavior.

The performance differed for the phosphazenic-type cocatalyst PPNN_3_ compared with that of the other two cocatalysts. Yields were above 80% for a polymerization time of 18 h for all three catalysts, where it reached a plateau. The same trend was observed for the molar masses and the glass transition temperatures, which both reached maximum values for 18 h of polymerization without further increases (Figure 3). A yield higher than 63% and an *M*_w_ of 9.8 kg/mol were obtained with the **2**/PPNN_3_ catalytic system for a polymerization time of 4 h (LOPA 33, Table 1). Unlike the other two cocatalysts, with PPNN_3_, the precontact step seemed to have no effect on the molar mass, which remained unchanged for precontact times of 1 and 24 h and for the same polymerization time (Figure 3b).

Figure 4 shows typical ^1^H NMR spectra of poly(LO-*alt*-PA) copolymers obtained with catalysts **1**–**3** and PPNCl as cocatalyst. The three spectra are similar and different from those of CHOPA and VCHOPA (vide infra). In addition, the ^1^H NMR spectra of LOPA do not show the typical signals of LO–LO ether linkage at 3.6 ppm.

As example the ^13^C NMR spectrum of a poly(LO-*alt*-PA) copolymer is reported (Appendix A).

To the best of our knowledge, the *T*_g_ values reported in Table 1 are the highest glass transition temperatures obtained for poly(LO-*alt*-PA) polyesters using commercial salen- type complexes [38]. The key to obtaining a high *M*_w_ and *T*_g_ is a precontact step between catalyst and cocatalyst for at least 1 h, as seen for poly(CHO-*co-*PA) polyesters [23]. Moreover, in the case of LO, the *T*_g_ can be modulated much more than with CHO and tuned from 104 to 136 °C, depending on the cocatalyst used and the precontact step duration.

Figure 5 presents the *T_g_* values as a function of molar mass (*M*_w_) for all poly(LO*-alt-*PA) samples. The trend in the curve consists of an increase up to a limit around the value of *M*_w_ = 11 kg/mol, followed by a plateau at which the increase in the *T*_g_ according to the molar mass is negligible.

Because biodegradability is one of the properties of these polymers, biodegradability tests were performed according to ISO standard 14851 [36] with the respirometric BOD (biochemical oxygen demand) method [39]. Figure 6 shows the average percentage of biodegradation plotted as a function of time for LOPA 08, LOPA 10, and the positive reference (microcrystalline cellulose). Moreover, the maximum level of biodegradation (MLB) detected within 60 days is reported in the inset of Figure 6. Biodegradation curves are typically characterized by: (i) a lag phase, from the start of the test until clear biodegradation; (ii) a biodegradation phase, where a sharp increase in the average percentage of biodegradation is recorded; (iii) a plateau phase, where biodegradation is almost completed. The degree of biodegradation of the reference sample (cellulose) at the end of the test exceeded the limit value of 60% required by the ISO 14851 to prove the validity of the test [36]. For the LOPA samples, the biodegradation phase was not obvious; for the cellulose, this phase started after 8 days. The curves of LOPA 08 and LOPA 10 almost overlapped, with a plateau phase close to 5%. For comparison purposes, the average percentage of biodegradation of a poly(CHO-*co*-PA) polyester [23] is displayed in Figure 6. The CHOPA sample showed higher biodegradation (>28%) than the LOPA samples due to a more evident susceptibility to attack by microorganisms. The cause of this difference probably lies in the nature of the chemical structure, a factor that mostly influences biodegradative processes. poly(LO-*alt*-PA) possesses substituent groups that limit accessibility to ester groups, on whose hydrolysis the degradation of the tested polymeric materials depends.

Finally, we proceeded with the functionalization of the vinyl double bond present in the poly(LO-*alt*-PA) copolymer chain using three different thiols (Figure 7) by a click chemistry reaction [35]. LOPA 98 (see Table 1) was modified with methyl-3-mercaptopropionate (M3MP), *iso*octyl-3-mercaptopropionate (I83MP), and butyl-3-mercaptopropionate (B3MP). The polymer prepared by the functionalization of LOPA 98 with thiols has been named LOPASH, and is reported in Table 2.

Functionalized LOPASH was prepared from LOPA 98. As shown in Table 2, yields higher than 55% and with variable functionalization percentages were obtained with all three thiols. The best results in terms of yield and functionalization percentage were obtained using the M3MP thiol. All obtained LOPASHs had molar masses similar to that of the starting LOPA 98 (*M*_n_ = 8.2 kg/mol), whereas the *T*_g_ values markedly change, ranging from 71 to 95 °C (Appendix A). The covalently attached butyl esters B3MP, M3MP, and I83MP considerably changed the thermal properties of the materials compared with those of the starting LOPA sample, with a noticeable decrease in the *T*_g_ of approximately 60 °C for the B3MP functionalized material.

Additionally, the thermal stability of pristine LOPA 98 and the three-thiol-modified poly(LO-*alt*-PA) was evaluated by TGA (Appendix A). The thermograms show that the thermal stability of LOPA material after thiol addition slightly decreased, with the initial degradation temperature (*T*_5%_ in Table 2) being about 5 °C lower than that of the starting LOPA sample. At higher temperatures, the LOPASH materials exhibited a fast mass loss with a maximum rate of degradation at approximately 270 °C, similar to the *T*_max_ value of the pristine poly(LO-*alt*-PA) copolymer. Figure 8 displays the ^1^H NMR spectrum of LOPASH 04 compared with that of LOPA 98.

In Figure 8, the spectra of LOPA98 and LOPASH04 are shown for comparison along with the repeating units of the two polyesters and the corresponding assignments of the protons in the spectra. The percentage of functionalization of double bonds in the LOPASH polyesters reported in Table 2 was calculated from the area of the signals **b** and **m** on the basis of the following Equation (3):(3)$$\%\;\mathrm{functionalization}=\frac{Area\;of\;signal\;m}{Area\;of\;signal\;b\;+\;Area\;of\;signal\;m}\times 100$$

### 3.2. ROCOP of Vinylcyclohexene Oxide and Phthalic Anhydride

We also tested the ROCOP of vinylcyclohexene oxide (VCHO), a monomer with a chemical structure similar to that of limonene oxide, except for the absence of the two methyl groups in positions 1 and 5, in the presence of PA. The polymerizations were performed using the same catalysts and cocatalysts used for the ROCOPs of limonene oxide with a precontact time of 24 h and a polymerization time of 1 h. The results are reported in Table 3. The polyesters obtained from ROCOP of VCHO and PA, poly(VCHOPA-*alt*-PA), are listed in Table 1 as VCHOPA.

In the presence of the chromium-based catalyst and regardless of the type of cocatalyst used, yields higher than 80% and molar masses ranging from 22.1 to 23.1 kg/mol with a distribution Đ of 1.2 were obtained. In the presence of both aluminum- and magnesium-based catalysts, polymerization yields were lower, and molar masses did not exceed 13.8 kg/mol with an average distribution Đ of 1.7.

The unimodal and bimodal molar mass distributions (MMDs) of poly(VCHO-*alt*-PA) synthesized in the presence of the different catalysts **1**, **2,** and **3** and PPNCl as cocatalysts are shown in Figure 9a. Unimodal poly(VCHO-*alt*-PA), VCHOPA 05 and 06, with molar masses (*M*_w_) of 13.8 and 10.8 kg/mol, respectively, and dispersity (Đ) of 1.6 were produced using catalysts **2** and **3**, respectively, after 1 h of polymerization. A bimodal MMD was observed for VCHOPA 01, 04, and 07 (Figure 9b). The higher molar mass fraction was nearly twofold higher than the lower one. The bimodal distribution is probably caused by hydrolyzed anhydride and/or traces of water acting as chain-transfer agents and causing diversification of the catalytic sites [30,40]. In addition, at high monomer conversions, side reactions such as transesterification and chain-end coupling reactions often occur in the ROCOP between cyclic anhydride and epoxides [31].

In Figure 10, the ^1^H NMR spectra of poly(VCHO-*alt*-PA) samples are compared. The magnification shows an area between 3.3 and 4.2 ppm where the peaks of the protons of the VCHO–VCHO ether linkage are present, along with those of the protons on the terminal monomeric units. The calculated amount of ether linkage slightly differs for the investigated poly(VCHO-*alt*-PA) samples, ranging from 1.5 to 3.0 mol%.

All poly(VCHO-*alt*-PA) samples were amorphous and showed the desired high glass transition temperature. We observed no major differences in the *T*_g_ value above 120 °C for all samples, with values up to 130 °C in the case of polymers with a higher molar mass obtained with the chromium-based catalyst (Appendix A).

The thermal stability and degradation mechanism of poly(VCHO-*alt*-PA) were investigated by TGA under an inert atmosphere, monitoring samples’ mass loss with increasing temperature. Figure 11a shows thermograms of the samples obtained from salen-type complexes with PPNCl as the cocatalyst as an example. Table 2 reports the TGA experimental data, including the onset of degradation, taken as the temperature at which 5% mass loss occurs (*T*_5%_), and the temperature of maximum rate of mass loss (*T*_max_).

Generally, the poly(VCHO-*alt*-PA) exhibited good thermal stability and similar thermal behavior, with a decomposition process that occurred in the temperature range between 300 and 500 °C. In detail, the poly(VCHO-*alt*-PA) samples showed onset decomposition temperatures at approximately 320 °C, except for the samples with a lower molar mass (VCHOPA 02 and VCHOPA 03, Table 3), which exhibited slightly lower *T*_5%_ values (304–307 °C). The main degradation step occurred between 300 and 400 °C, with a corresponding DTG peak centered at ca. 365 °C, which was associated with approximately 95% of the mass loss for each sample.

Notably, poly(VCHO-*alt*-PA) showed higher thermal stability than poly(LO-*alt*-PA). In Figure 11b, the thermograms of LOPA79 and VCHOPA 06, polyesters with a similar molar mass and distribution Đ, are compared. VCHOPA 06 showed higher *T*_5%_ and *T*_max_ values (327 and 367 °C, respectively) than LOPA 79 (263 and 282 °C, respectively). A comparable thermal behavior with low decomposition onsets spanning from 234 to 258 °C was reported for poly(LO-*alt*-PA) polyesters prepared by different catalytic systems [27,41]. This different thermal behavior is probably due to the presence of weak links, such as methyl branches, along the backbone LOPA copolymer chain. For poly(LO-*alt*-PA), the presence of a quaternary carbon atom weakens the C–O bond and therefore facilitates the degradation process.

For comparison purposes, the thermal stability of a poly(CHO-*co*-PA) polyester with *M*_w_ = 16 kg/mol and Đ = 1.1 is displayed in Figure 11b. The thermograms of VCHOPA 06 and CHOPA polyesters, which have similar chemical structure except for the presence of a vinyl group in position 4, are almost overlapping. This confirmed that the chemical structure is the key factor determining the thermal stability of the investigated polyesters.

Importantly, all the polyesters described in Table 1 and Table 3 exhibited decomposition onset temperatures higher than their respective *T*_g_ values (typically 110−130 °C for poly(LO-*alt*-PA) and 190−200 °C for poly(VCHO-*alt*-PA)), which enable a wide temperature window for processing these polyesters. Some preliminary compression molding tests were successfully conducted at 170 °C and 50 bar for 5 min (Figure 12).

## 4. Conclusions

Here, we presented the synthetic route for preparing aliphatic polyesters with high molar masses and high glass transition temperatures starting from limonene oxide, a suitable green platform molecule.

The effects of precontact time between the catalyst and cocatalyst and polymerization time on the productivity, molar mass, and glass transition temperature were examined. The time of the precontact step positively affected the polymerization yield and molar mass of the polyesters by using **1**–**3** with DMAP and PPNCl as the cocatalysts. For both DMAP and PPNCl cocatalysts, as the reaction time increased, the molar masses gradually increased with a narrow molar mass distribution (ca. 1.2), indicating living polymerization behavior. Otherwise, for PPNN_3,_ the precontact time seemed to have no effect on the molar mass, which remained unchanged for a precontact step of 1 or 24 h and for the same polymerization time.

For poly(LO-*alt*-PA) copolymers, biodegradability tests were performed. The maximum biodegradation was much lower than that of poly(CHO-*co*-PA) copolymers obtained by ROCOP of CHO with PA. This different biodegradation behavior is probably due to the chemical structure of poly(LO-*alt*-PA), which possesses methyl substituent groups that limit the accessibility to the ester groups, on whose hydrolysis the degradation of the tested polymeric materials depends. The vinyl double bonds functionalization of poly(LO-*alt*-PA) copolymer chains allowed the tuning of the thermal properties of these aliphatic polyesters, keeping the molar mass unchanged.

Poly(VCHO-*alt*-PA) copolymers were also synthetized with **1**–**3** and DMAP, PPNCl, and PPNN_3_ as cocatalysts. The polyesters were characterized by a molar mass of up to 23.0 kg/mol and a narrow distribution, a glass transition temperature up to 130 °C, and a low content (<3 mol%) of polyether units.

All the alternating polyesters exhibited decomposition onset temperatures higher than their respective *T*_g_ values, providing a wide temperature window for processing these bio-based polyesters.

## Data Availability

Data are contained within the article or Appendix A.

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
