# Peer review of "High-Glass-Transition Polyesters Produced with Phthalic Anhydride and Epoxides by Ring-Opening Copolymerization (ROCOP)"

_polymers, 2023, doi:10.3390/polym15132801_

Round 1
Reviewer 1 Report
Comments for authors on manuscript polymers-2418096: "High glass transition polyesters by phthalic anhydride and epoxides by Ring-Opening COPolymerization (ROCOP)” by S. Silvano, Matteo Proverbio, A. Vignali, F. Bertini and L. Boggioni.
In this manuscript the authors describe the synthesis of different polyesters from limonene oxide or vinylcyclohexene oxide and phthalic anhydride (PA) by ROCOP using commercial SALEN complexes (Cr, Al, and Mn) and different cocatalysts as system catalytics. This report was carefully performed and included several interesting observations, a significant advance could find from present report. The study of the degradation of biodegradable polymers such as the polyesters is a very attractive field for the scientific community and for the industry, therefore the manuscript is suitable for publication in Polymers after minor revision:
- The authors must improve the presentation of the manuscript, 1H-NMR and 13C{1H}-NMR spectra of all polymers should be added in the supplementary material to facilitate follow-up of comments made by the authors.
- The mass spectra of the polymers should be added, these would allow studying the end groups of the chain and understanding which is the catalytically active species.
- The authors comment on the importance of the precontact stage between the catalyst and the cocatalyst for the properties of the polyesters and for the catalytic activity of the compounds supported by the SALEN ligands. In my opinion the authors should study this reaction in more depth and observe what happens in the catalytic system when the catalyst and the cocatalyst come into contact, this must be explained to draw these conclusions.
- The authors must carry out tests with the cocatalyst only to verify the activity of these compounds in the process. The same with the catalyst.
- The authors should improve the explanation of the functionalization processes of the polymers, they comment on a click chemistry process, and nothing appears in figure 7. To improve the understanding of the manuscript, a scheme where the functionalization of the polymers is collected would be well received.
Reviewer 2 Report
In this manuscript (polymers-2418096), Drs. Bertini, Boggioni, and co-authors studied the preparation of high glass transition temperature (Tg) polyesters through ring-opening copolymerization (ROCOP) with phthalic anhydride and bio-based epoxides. Three salen-type complexes with metals as catalysts and DMAP, PPNCl, and PPNN3 as cocatalysts were applied in the ROCOP to investigate the effects of precontact time between catalysts and cocatalysts and polymerization time on the productivity, molar mass, and Tg. It was found that the polyesters obtained via the ROCOP in this study featured molar mass up to 14 kg/mol, narrow dispersity, and higher Tg as well as low polyether units. Through a click (thiol-ene) reaction between the thiol and vinyl group, the poly(LO-alt-PA) was further functionalized with three different mercapto compounds. The thermal stability and biodegradation of these polyesters were also evaluated.
It is important to develop novel polyesters with good thermal properties as well as biodegradability. It would appeal to the broad readership of Polymers, especially those who focus on studying bio-based polyesters. However, some of the key features in the manuscript were missed, and the language should be thoroughly improved. Based on those, major revisions are needed before further consideration.
1. It would make it more readable if the role of these salen-type complexes with metals is clearly stated as a catalyst in the Abstract section.
2. The authors should keep it consistent when writing the unit for polymer molar mass. In some places, it was written as kg/mol, while in other places it was written as Kg/mol, like on lines 16 and 63, respectively.
3. The name of the chemical on line 58 should be “1,4-cyclohexadiene oxide”.
4. The sentence on line 69 was not correct. It is a fragment if written as “By tuning, properly, the anhydride…”. The comma should be used properly.
5. It was noticed that in some parts (like on line 253), the authors make Mw as molar mass for polymers. However, it is treated as weight mass for Mw. The molar mass should be Mn, and they are two different definitions for clarifying the mass of polymers. The authors should correct them and avoid misleading the readers in this aspect.
6. The letters (N and N’) and words (tert and iso) should be italicized when used in chemical nomenclature (section 2.1).
7. Detailed conditions or methods (like eluting through a basic Al gel column) should be provided for the pre-purification of all reagents used for polymerization on line 192.
8. What does the “CHO e PA” mean on line 197?
9. The authors should explain what the number means in the column for “Entry” in Table 1, like “LOPA 07”.
10. It is hard to understand the sentence “… with all and three catalytic systems…” on lines 209-210. What does it mean “all and three catalytic systems”? Or what are they?
11. On lines 215 and 221, “at shorter times” should be revised as “at a shorter time”.
12. What are the solvents used for the proton NMR tests in Fig 4? It should be provided in the legend of the figure.
13. Based on the meaning of what the authors wanted to convey, it should be “the highest” instead of "the higher” on line 247. Otherwise, it is also a grammatically wrong sentence.
14. It is not proper to write “…, CHOPA 19 in [23], …” on line 276, as it makes no sense and would confuse the readers as we don’t know what it means.
15. What would be the reason(s) that CHOPA 19 shows a sudden increase in biodegradation between days 30 to 35 in Fig 6?
16. A scheme showing the structures of the polyesters prepared in this report should be provided. Otherwise, it would be difficult to imagine where is “position 4” in the polyesters on lines 388-389. It also weakens the strength of the first sentence in the Conclusions section.
17. Figs S1 and S3 are recommended to show the full circles of DSC tests. An inserted section showing the Tg area can be used to zoom in on the regions where further discussion is given.
18. A review paper (https://doi.org/10.1002/marc.202100054) would be used as a reference to discuss the functional polyesters in degradation for the Introduction section.
The language needs to be improved thoroughly.
Reviewer 3 Report
Boggioni et al. reported in this paper preparation of high glass transition polyesters by ring-opening copolymerization (ROCOP) of phthalic anhydride and epoxides. Optimization of the synthetic conditions were performed, and physical properties of the synthesized polyesters were investigated. The acquired useful data merit the publication in Polymers. The following minor concerns should be addressed prior to acceptance.
1. The uniqueness of this study relative to the published papers with a similar research topic on the use of ROCOP to produce polyesters with high Tg remain unclear, which should be highlighted clearly in the abstract and introduction sections.
2. Pseudo-first-order kinetics of the ROCOP process should be investigated and reported to confirm the living characteristics.
3. Typical SEC elution traces of the synthesized polymers should be provided in the main text.
4. The characteristic proton signals in the NMR spectra (Figure 9) should be assigned together with the chemical structures of the synthesized polyesters.
The language is qualified for publication.
Round 2
Reviewer 2 Report
In the revised manuscript, the authors have discussed and provided detailed explanations to the questions from the reviewers. The quality of the manuscript has been improved evidently. One minor revision is still needed as on line 159, the ref 35 was not correctly cited. Please change it before publication.
Author Response
The authors thank reviewer 2 for careful editing of the manuscript.
Reference 35 was corrected on line 159.